

# Dual network embedding for representing research interests in the link prediction problem on co-authorship networks

Ilya Makarov[1,2], Olga Gerasimova[1], Pavel Sulimov[1] and Leonid E. Zhukov[1]

[1] School of Data Analysis and Artificial Intelligence, National Research University Higher School of Economics, Moscow, Russia
[2] Faculty of Computer and Information Science, University of Ljubljana, Ljubljana, Slovenia

## ABSTRACT

We present a study on co-authorship network representation based on network embedding together with additional information on topic modeling of research papers and new edge embedding operator. We use the link prediction (LP) model for constructing a recommender system for searching collaborators with similar research interests. Extracting topics for each paper, we construct keywords co-occurrence network and use its embedding for further generalizing author attributes. Standard graph feature engineering and network embedding methods were combined for constructing co-author recommender system formulated as LP problem and prediction of future graph structure. We evaluate our survey on the dataset containing temporal information on National Research University Higher School of Economics over 25 years of research articles indexed in Russian Science Citation Index and Scopus. Our model of network representation shows better performance for stated binary classification tasks on several co-authorship networks.

## INTRODUCTION

Nowadays, researchers struggle to find relevant scientific contributions among large variety of international conferences and journal articles. In order not to miss important improvements in various related fields of study, it is important to know the current state-of-art results while not reading all the papers tagged by research interests. One of the solutions is to search for the most "important" articles taking into account citation or centrality metrics of the paper and the authors with high influence on specific research field (*Liang, Li & Qian, 2011*). However, such method does not include collaborative patterns and previous history of research publications in co-authorship. It also does not measure the author professional skills and the ability to publish research results according to paper influence metrics, for example, journal impact factor.

We study the problem of finding collaborator depending on his/her research community, the quality of publications and structural patterns based on co-authorship network suggested by *Newman (2004a, 2004b)*. Early unsupervised learning approaches

Corresponding author
Ilya Makarov, iamakarov@hse.ru

for community detection in research networks were studied in *Morel et al. (2009)*, *Cetorelli & Peristiani (2013)*, *Yan & Ding (2009)*, and *Velden & Lagoze (2009)*. A review on social network analysis and network science can be found in *Wasserman & Faust (1994)*, *Barabási & Pósfai (2016)*, and *Scott (2017)*.

We focus on the link prediction (LP) problem (*Liben-Nowell & Kleinberg, 2007*) in order to predict links in temporal networks and restore missing edges in complex networks constructed over noisy data. The LP algorithms can be used to extract the missing link or to detect abnormal interactions in a given graph, however, the most suitable case is to use LP for predicting the most probable persons for future collaboration, which we state as a problem of recommending a co-author using LP ranking (*Li & Chen, 2009*). Our model is designed to predict whether a pair of nodes in a network would have a connection. We can also predict the parameters of such an edge in terms of publication quality or number of collaborators corresponding to the predicted link (*Makarov et al., 2019a*, *2019b*).

In general, LP algorithms are widely used in several applications, such as web linking (*Adafre & De Rijke, 2005*), search for real-world friends on social networks (*Backstrom & Leskovec, 2011*), citation recommender system for digital libraries (*He et al., 2010*). A complete list of existing applied LP techniques can be found in *Srinivas & Mitra (2016)*.

Recently, the improvement of machine learning techniques shifted the attention from manual feature engineering to the vectorized information representation. Such methods have been successfully applied for natural language processing and now are tested on network topology representation despite the fact that an arbitrary graph could not be described by its invariants. The approach of representing network vertices by a vector model depending on actor's neighborhood and similar actors is called graph (network) embedding (*Perozzi, Al-Rfou & Skiena, 2014*). The current progress of theoretical and practical results on network embeddings (*Perozzi, Al-Rfou & Skiena, 2014*; *Tang et al., 2015*; *Chang et al., 2015*; *Grover & Leskovec, 2016*) shows state-of-art performance on such problems as multi-class actor classification and LP. Although, the existing methods use not only structural equivalence and network homophily properties, but also the actor attributes, such as labels, texts, images, etc. A list of surveys on graph embedding models and applications can be found in *Cai, Zheng & Chang (2018)*, *Cui et al. (2018)*, *Goyal & Ferrara (2018)*, and *Chen et al. (2018)*.

In this paper, we study a co-authorship recommender system based on co-authorship network where one or more of the coauthors belong to the National Research University Higher School of Economics (NRU HSE) and the co-authored publications are only those indexed in Scopus. We use machine learning techniques to predict new edges based on network embeddings (*Grover & Leskovec, 2016*; *Wu & Lerman, 2017*) and edge characteristics obtained from author attributes. We compare our approach with state-of-the-art algorithms for the LP problem using structural, attribute and combined feature space to evaluate the impact of the suggested approach on the binary classification task of predicting links in co-authorship network. Such an obtained system could be applied for expert search, recommending collaborator or scientific adviser, and searching for relevant research publications similar to the work proposed in

*Makarov, Bulanov & Zhukov (2017)* and *Makarov et al. (2018a)*. In what follows, we describe solution to the LP problem leading to evaluation of our recommender system based on co-authorship network embeddings and manually engineered features for HSE researchers.

## RELATED WORK

### Link prediction

The LP problem was stated in *Liben-Nowell & Kleinberg (2007)*, in which Liben-Nowell and Kleinberg proposed using node proximity metrics. The evaluation of the proposed metrics for large co-authorship networks showed promising results for predicting future links based on network topology without any additional information on authors. Unsupervised structural learning was proposed in *Tang & Liu (2012)*. *Gao, Denoyer & Gallinari (2011)* presented temporal LP based on node proximity and its attributes determined by the content using matrix factorization.

Two surveys on LP methods describe core approaches for feature engineering, Bayesian approach and dimensionality reduction were presented in *Hasan & Zaki (2011)*, *Lü & Zhou (2011)*. Survey on LP was published in *Wang et al. (2015)*.

The simplest baseline solution using network homophily is based on common neighbors or other network similarity scores (*Liben-Nowell & Kleinberg, 2007*). However, the *Gao et al. (2015)* that the similarity measures are not robust to the network global properties and, thus, could noise the prediction model with similarity scores only. The impact of the attribute-based formation in social networks was considered in *Robins et al. (2007)* and *McPherson, Smith-Lovin & Cook (2001)*. All these observations require feature engineering depending on the domain.

Graph-based recommender systems formulated via LP problem were suggested in *Chen, Li & Huang (2005)*, *Liu & Kou (2007)*, and *Li & Chen (2009)*. In *Kossinets & Watts (2009)*, studied the effect of homophily in a university community. They considered temporal co-authorship network accompanied with author attributes and concluded the influence of not only structural proximity, but also author homophily for the social network structure.

Another approach focusing on interdisciplinary collaboration inside the University was presented in *Cho & Yu (2018)*. The authors used the existing co-authorship network and academic information for University of Bristoll and proposed a new LP model for co-authorship prediction and recommendation.

*Kong et al. (2018)* developed a scientific paper recommender system called VOPRec. However, in contrast to our work they constructed vector representation of research papers in citation networks. Their system uses both text information represented with word embedding to find papers of similar research interest and structural identity converted into vectors to find papers of similar network topology. To combine text and structural informations with the network, vector representation of article can be learned with network embedding.

### Network embedding

In general, knowledge retrieval and task-dependent feature extraction would require domain-specific expert to construct a real-value feature vector for nodes and edges

representation. The quality of such an approach will be influenced by particular tasks and expert work, while not being scalable for large noisy networks. Recently, the theory of hidden representations has impacted on machine learning and artificial intelligence. It shifted the attention from manual feature engineering to defining loss function and then solving optimization task. The early works on network vectorized models were presented in Local Linear Embedding (*Roweis & Saul, 2000*), IsoMAP (*Tenenbaum, De Silva & Langford, 2000*), Laplacian Eigenmap (*Belkin & Niyogi, 2002*), Spectral Clustering (*Tang & Liu, 2011*), MFA (*Yan et al., 2007*), and GraRep (*Cao, Lu & Xu, 2015*). These works try to embed the networks into real-value vector space using several proximity metrics. However, development of representation learning for networks was in stagnation due to the non-robust and non-efficient machine learning methods of dimensionality reduction based on network matrix factorization or spectral decomposition. These methods were not applicable for large networks and noisy edge and attribute data providing low accuracy and having high time complexity of constructing embedding.

The modern methods of network embedding try to improve the performance on several typical machine learning tasks using conditional representation of a node based on its local and global neighborhood defined via random walking. The first-order and second-order nodes proximity were suggested in LINE (*Tang et al., 2015*) and SDNE (*Wang, Cui & Zhu, 2016*) models. Generalizing this approach, DeepWalk (*Perozzi, Al-Rfou & Skiena, 2014*) and node2vec (*Grover & Leskovec, 2016*) algorithms use Skip-gram model (*Mikolov et al., 2013*) based on simulation of breadth-first sampling and depth-first sampling. Although, in *Carstens et al. (2017)*, Carstens et al. showed some drawbacks of node2vec (*Grover & Leskovec, 2016*) graph embedding, it still remained competitive structural-only embedding for representing both, homophily and structural equivalence in the network. Its generalization on global network representation learning from *Wu & Lerman (2017)* shows comparable results with the original model.

Several works cover the node attributes, such as label and text content (see TADW; *Yang et al., 2015*, LANE; *Huang, Li & Hu, 2017*). In TriDNR paper, *Pan et al. (2016)* proposed to separately learn structural embedding from DeepWalk (*Perozzi, Al-Rfou & Skiena, 2014*) and content embedding via Doc2Vec (*Le & Mikolov, 2014*). On the contrary, ASNE (*Liao et al., 2017*) learns combined representations for structural and node attribute representation using end-to-end neural network.

We focus on graph embedding and feature engineering methods applied to an LP task on a particular network, consisting of HSE researchers co-authored at least one paper with additional attributes representing authors. Using network features only fails to include the information about actors obtained from the other sources, thus decreasing efficiency of network embeddings. We aim to include information on feature space of author's research interests using data from the Scopus digital library containing manually input and automatically selected keywords for each research article. Based on this information, we constructed keywords co-occurrence network and consider its embedding for further generalizing author attributes.

# DATASET DESCRIPTION AND PREPROCESSING

We use the NRU HSE portal (*National Research University Higher School of Economics, 2017*) containing information on research papers co-authored by at least one HSE researcher, which were later uploaded to the portal by one of co-authors. The HSE database contains information on over 7,000 HSE researchers published over 31,000 research papers. The portal site contains web interface for the researchers to extract metadata of publications for a given time period and could be used by external researchers. The database records contain information on title, list of authors, keywords, abstract, year and place, journal/conference and publishing agency, and indexing flags for Scopus, Web of Science (WoS) Core Collection and Russian Science Citation Index (RSCI).

Unfortunately, the database has no interface for managing bibliography databases and has no integration with synchronizing of indexing digital libraries compared to Scholar Google or personal researcher profile management services such as ResearcherID or Orcid. As a consequence, a large amount of noisy data occurs due to such problems as author name ambiguity or incorrect/incomplete information on the publications.

In order to resolve the ambiguity, we considered standard disambiguation approaches for predicting necessity to merge authors. We used Levenshtein distance (useful for records with one-two error letters) for abbreviated author last and first names and then validated by two thresholds whether to merge two authors with the same abbreviation in the database based on cosine similarity and common neighbors metrics. The threshold values have been found manually via validation on small labeled set of ambiguous records. The number of authors with ambiguous writing does not exceed 2% of the whole database. We have also removed all non-HSE authors due to lack of information on their publications in HSE dataset.

We also retrieved the Scopus database of research papers co-authored by researchers from NRU HSE and indexed by *Elsevier (2018)*. The database contains information on paper author list, document title, year, source title, volume, issue, pages, source and document type, DOI, author keywords, index keywords. We also added the information on research interests based on Scopus subject categories for the journals, in which authors have published their articles. We manually inputted the research interest list according to RSCI categorization in order to fill the lack of keywords and attributes for the papers.

We then stated the problem of indexing author research interests in terms of keywords attached to paper description in both databases, and retrieved from HSE dataset using the BigARTM (*Vorontsov et al., 2016*) topic modeling framework. For the Scopus dataset, we use automatically chosen keywords previously prepared by the service together with manually input by authors list of keywords. We also uses additional keywords written in terms of subject categories of journals and proceedings according to the indexing in Scopus and WoS research paper libraries.

These two datasets (HSE, Scopus) have common papers; however HSE dataset contains many noisy data and, unfortunately, low-level publications not indexed outside RSCI, while Scopus contains precise information on 25% number of papers and exact research

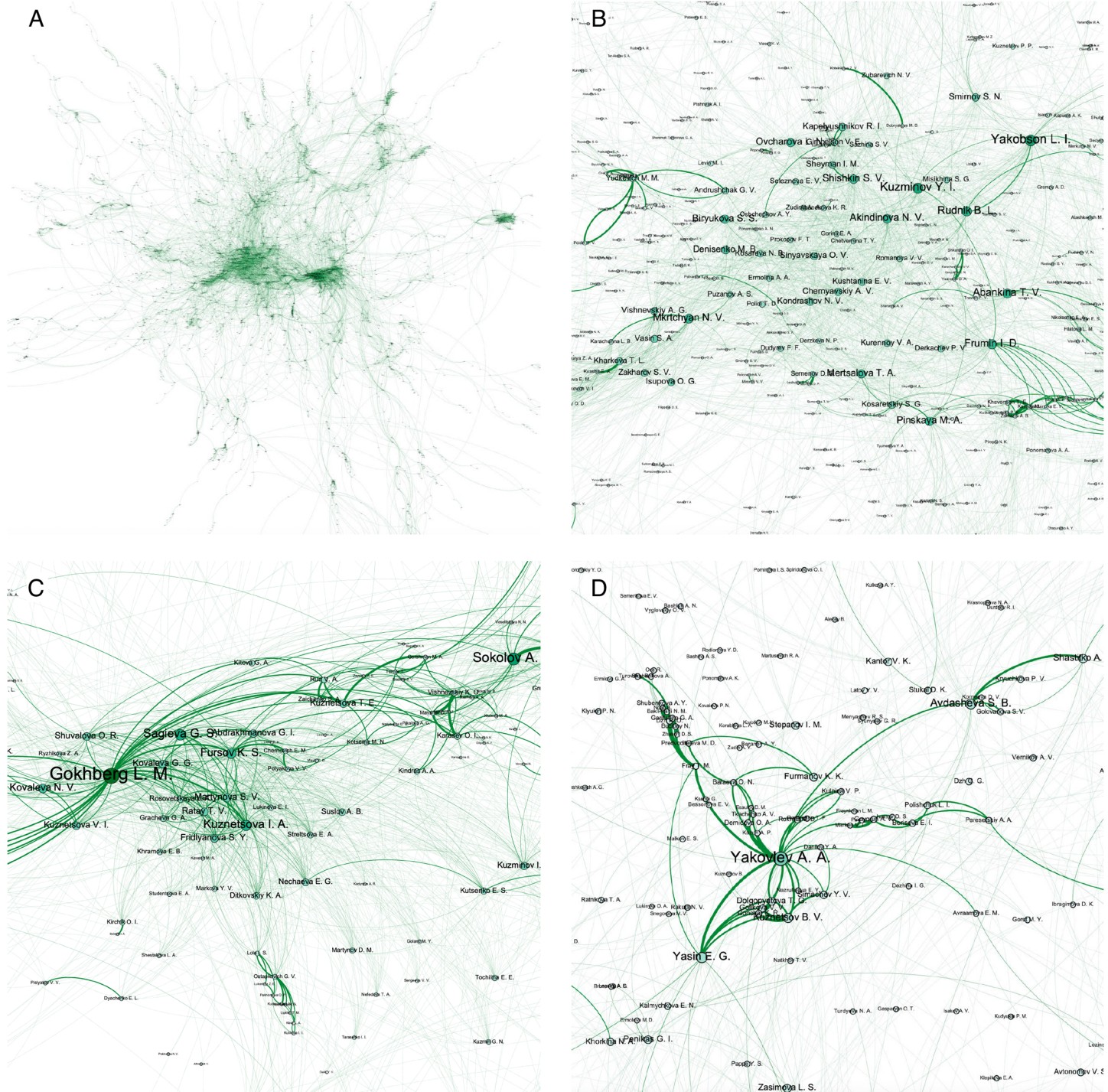

**Figure 1 Visualization of HSE co-authorship network.** We plot the whole HSE co-authorship network (A) and its subgraphs induced by local proximities around influential persons from our university such as rector Kuzminov Y.I. (B), first vice-rector responsible for science Gokhberg L.M. (C), and university research supervisor Yasin E.G. (D).

interest representation while lacking weak connections on authors written only poor-quality papers.

We visualized the HSE network with author names as node labels shown in the Fig. 1 while visualizing edge width as the cumulative quantity of joint publications based on the

papers' quartiles and their number. In particular, we plotted the whole network structure (Fig. 1A) and zoomed parts corresponding dense communities with the most influential persons from our university such as rector Kuzminov Y.I. (Fig. 1B), first vice-rector responsible for science Gokhberg L.M. (Fig. 1C), and university research supervisor Yasin E.G. (Fig. 1D). It is easy to see that rector co-authors are people responsible for core direction of university development and vice rectors realizing university strategy in research, education, government, expert, and company areas of collaboration. On the other hand, from dense subgraphs one can find exact matches of university staff departments, such as research institutes, such as Institute for Industrial and Market Studies headed by Yakovlev A.A. or Institute for Statistical Studies and Economics of Knowledge headed by L.M. Gokhberg.

We show that network could visualize the most important administrative staff units and their heads thus giving insight on the connection of publication activity and administrative structure of HSE university.

## FEATURE ENGINEERING

Our new idea was to obtain additional edge attributes that were embed based on keywords network as a part of model evaluation. We constructed the network of stemmed keywords co-occurrence. To construct this network, we used the principle that two nodes were connected if corresponding keywords occurred in the same paper. For a given list of keywords, we built standard node2vec embedding (Grover & Leskovec, 2016). Next, for each author the most frequent and relevant keyword was defined, and its embedding was used as node additional feature vector for our LP tasks.

We considered the problem of finding authors with similar interests to a selected one as collaboration search problem. In terms of social network analysis, we studied the problem of recommending similar author as LP problem. We operate with authors similarity and use similarity scores described in Liben-Nowell & Kleinberg (2007) as baseline for network descriptors for pairs of nodes presented in Table 1.

So, we represented each node by vector model of author attributes using manually engineered features such as HSE staff information and publication activity represented by centralities of co-authorship network and descriptive statistics. We added graph embeddings for author research interests and node proximity and evaluated different combinations of models corresponding to node feature space representation.

## LINK EMBEDDINGS

To use node2vec, we obtained the vector node representations. The node2vec embedding parameters were chosen via ROC–area under the curve (AUC) optimization over embedding size with respect to different edge embedding operators. For edge embedding we applied specific component-wise functions representing edge to node embeddings for source and target nodes of a given edge. This model was suggested in Grover & Leskovec (2016), in which four functions for such edge embeddings were presented (see first four rows in Table 2). We leave an evaluation of the approaches from Abu-El-Haija, Perozzi & Al-Rfou (2017), which use bi-linear form learning from reduced by deep neural

**Table 1 Similarity score for a pair of nodes $u$ and $v$ with local neighborhoods $N(u)$ and $N(v)$ correspondingly, and for vectors corresponding to two authors research interests $X$ and $Y$.**

| Similarity metric | Definition |
|---|---|
| Common neighbors | $|N(u) \cap N(v)|$ |
| Jaccard coefficient | $\dfrac{|N(u) \cap N(v)|}{|N(u) \cup N(v)|}$ |
| Adamic-Adar score | $\sum_{w \in N(u) \cap N(v)} \dfrac{1}{\ln |N(w)|}$ |
| Preferential attachment | $|N(u)| \cdot |N(v)|$ |
| Graph distance | Length of shortest path between $u$ and $v$ |
| Metric score | $\dfrac{1}{1 + ||x - y||}$ |
| Cosine score | $\dfrac{(x, y)}{||x|| \, ||y||}$ |
| Pearson coefficient | $\dfrac{\mathrm{cov}(x, y)}{\sqrt{\mathrm{cov}(x, x)} \cdot \sqrt{\mathrm{cov}(y, y)}}$ |
| Generalized Jaccard | $\dfrac{\sum \min(x_i, y_i)}{\sum \max(x_i, y_i)}$ |

**Table 2 Binary operators for computing vectorized $(u, v)$-edge representation based on node attribute embeddings $f(x)$ for $i$th component for $f(u, v)$.**

| Symmetry operator | Definition |
|---|---|
| Average | $\dfrac{f_i(u) + f_i(v)}{2}$ |
| Hadamard | $f_i(u) \cdot f_i(v)$ |
| Weighted-$L_1$ | $|f_i(u) - f_i(v)|$ |
| Weighted-$L_2$ | $(f_i(u) - f_i(v))^2$ |
| Neighbor Weighted-$L_1$ | $\left| \dfrac{\sum_{w \in N(u) \cup \{u\}} f_i(w)}{|N(u)| + 1} - \dfrac{\sum_{t \in N(v) \cup \{v\}} f_i(t)}{|N(v)| + 1} \right|$ |
| Neighbor Weighted-$L_2$ | $\left( \dfrac{\sum_{w \in N(u) \cup \{u\}} f_i(w)}{|N(u)| + 1} - \dfrac{\sum_{t \in N(v) \cup \{v\}} f_i(t)}{|N(v)| + 1} \right)^2$ |

network node embedding, for the future work while also working on a new model of joint node-edge graph embedding, similar to (*Goyal et al., 2018*). Presented in the paper model suggests simple generalization of the idea of pooling first-order neighborhood of nodes while constructing edge embedding operator, which is much faster than dimensionality reduction approaches.

We evaluated our model on two additional functions involving not only edge source and target node representations, but also their neighborhood representations as average over all the nodes in first-order proximity. These measures were first presented in *Makarov et al. (2018b)* but were not properly evaluated. The resulting list of link embeddings is presented in Table 2.

For each author, we also chose the most frequent keyword in the co-occurrence network and then constructed general embedding using node2vec with automatically chosen parameters.

Overall edge embedding contained several feature spaces described by the following List 1:

(1) Edge embedding based on node2vec node embeddings

(2) Edge embedding based on node2vec embedding for keywords co-occurrence network

(3) Network similarity scores (baselines in *Liben-Nowell & Kleinberg (2007)*)

- Common neighbors
- Jaccard's coefficient
- Adamic-Adar score
- Preferential attachment
- Graph distance

(4) Author similarity scores

- Cosine similarity
- Common neighbors
- Jaccard's generalized coefficient
- Pearson's correlation coefficient
- Metric score

## TRAINING MODEL

We consider machine learning models for binary classification task of whether for a given pair of nodes there will be a link connecting them based on previous or current co-authorship network. We compare Logistic Regression with Lasso regularization, Random Forest, Extreme Gradient Boosting (XGBoost), and Support Vector Machine, in short SVM, models.

We use most common machine learning frameworks which we shortly describe below. *Logistic Lasso Regression* is a kind of linear model with a logit target function, in addition, Lasso regularization sets some coefficients to zero, effectively choosing a simpler model that reduces number of coefficients. *Random Forest* and *Gradient Boosting* are an ensemble learning methods. The former operates by constructing a multitude of decision trees, the latter combines several weak models building the model in a stage-wise fashion and generalizes them by allowing optimization of an arbitrary differentiable loss function. Random Forest adds additional randomness to the model, while growing the trees. Instead of searching for the most important feature while splitting a node, it searches for the best feature among a random subset of features. *SVM* constructs a hyperplane that has the largest distance to the nearest training-data point of any class when it achieves a good classification.

We use standard classification performance metrics for evaluating quality such as Precision, Accuracy, F1-score (micro, macro), Log-loss, ROC–AUC. Next, we shortly define them.

*Precision* is a measure that tells us what proportion of publications that we diagnosed as existing, actually had existed. *Accuracy* in classification problems is the number of correct

**Table 3 Comparing machine learning models based on the Neighbor Weighted-$L_2$ link embedding applied to future links prediction on the Scopus dataset.**

|  | Precision | Accuracy | F1-score (macro) | F1-score (micro) | Log-loss | ROC–AUC |
|---|---|---|---|---|---|---|
| Logistic regression | 0.382 | 0.372 | 0.483 | 0.471 | 2.089 | 0.534 |
| Random forest | 0.622 | 0.671 | 0.652 | 0.641 | 0.893 | 0.725 |
| Gradient boosting | 0.487 | 0.521 | 0.527 | 0.582 | 1.275 | 0.631 |
| SVM | **0.712** | **0.784** | **0.761** | **0.754** | **0.634** | **0.816** |

predictions made by the model over all kinds predictions made. *Recall* is a measure that tells us what proportion of publications that actually had existed was diagnosed by the algorithm as existing. To balance these metrics, F1-score is calculated as a weighted mean of the Precision and Recall. *Micro-averaged* metrics are usually more useful if the classes distribution is uneven by calculating metrics globally to count true and false predictions, *macro-averaged* are used when we want to evaluate systems performance across on different classes by calculating metrics for each label. *Log-loss*, or logarithmic loss, is a "soft" measurement of accuracy that integrates the idea of probabilistic confidence. In binary classification Log-loss can be calculated as $-\frac{1}{N} \cdot \sum_{i=1}^{N}(y_i \cdot \log(p_i) + (1 - y_i) \cdot \log(1 - p_i))$, where $y_i$ is a binary indicator (0 or 1) checking the correctness of classification for observation $i$, $p_i$ is a predicted probability observation $i$ for a given class. Log-loss measures the unpredictability of the "extra noise" that comes from using a predictor as opposed to the true labels. ROC–AUC, aka area under the receiver operating characteristic curve, is equal to the probability that a classifier will rank a randomly chosen existing edge higher than a randomly chosen non-existing one. ROC curves typically feature rate of true predicted existing publications on the $Y$ axis, and rate of false predicted existing publications on the $X$ axis. The larger AUC is usually better. All metrics were averaged using fivefold cross validation with five negative sampling trials for a fixed train set, which we describe below.

In our LP problem for the co-authorship network, we have two possible formalizations of predicting links. We consider either temporal network structure using information from the previous years to predict links corresponding the current year or the whole network to predict missing links. For the first task, we use combined HSE+Scopus dataset of 2015 publications and learn to predict papers appearing at 2016 year. We test our model shifting years on 1 year ahead and evaluate our predictions for 2017 year based on publications until 2016 year. For the second task, we remove 50% of existing edges preserving connectivity property from our dataset and add negative sampling of the same size as a number of edges left in order to balance classes for classification problem.

In Table 3 for the future links prediction task, we compare chosen predictive models fixing one Neighbor Weighted-$L_2$ link operator to construct edge embeddings considered as model features. It is interesting to see that XGBoost model get significantly overfitted while the best model appear to be Support Vector Machine. In Table 3 and in further tables we highlight the best values of quality metrics in bold.

**Table 4 Comparing link embeddings for future links prediction on the Scopus dataset.**

|  | Precision | Accuracy | F1-score (macro) | F1-score (micro) | Log-loss | ROC–AUC |
|---|---|---|---|---|---|---|
| Average | 0.436 | 0.578 | 0.661 | 0.589 | 1.341 | 0.599 |
| Hadamard | 0.613 | 0.654 | 0.657 | 0.628 | 1.125 | 0.634 |
| Weighted-$L_1$ | 0.645 | 0.678 | 0.674 | 0.632 | 0.979 | 0.723 |
| Weighted-$L_2$ | 0.672 | 0.682 | 0.688 | 0.637 | 0.915 | 0.742 |
| Neighbor Weighted-$L_1$ | 0.644 | 0.692 | 0.701 | 0.696 | 0.832 | 0.783 |
| Neighbor Weighted-$L_2$ | **0.712** | **0.784** | **0.761** | **0.754** | **0.634** | **0.816** |

**Table 5 Comparing machine learning models based on the Neighbor Weighted-$L_2$ link embedding for link prediction problem on the HSE dataset.**

|  | Precision | Accuracy | F1-score (macro) | F1-score (micro) | Log-loss | ROC–AUC |
|---|---|---|---|---|---|---|
| Logistic regression | 0.421 | 0.462 | 0.502 | 0.472 | 1.873 | 0.583 |
| Random forest | 0.836 | 0.871 | 0.774 | 0.831 | 0.193 | 0.888 |
| Gradient boosting | 0.771 | 0.732 | 0.703 | 0.734 | 0.742 | 0.663 |
| SVM | **0.823** | **0.845** | **0.782** | **0.812** | **0.273** | **0.828** |

**Table 6 Comparing machine learning models based on the Neighbor Weighted-$L_2$ link embedding for link prediction problem on the Scopus dataset.**

|  | Precision | Accuracy | F1-score (macro) | F1-score (micro) | Log-loss | ROC–AUC |
|---|---|---|---|---|---|---|
| Logistic regression | 0.482 | 0.491 | 0.522 | 0.563 | 1.452 | 0.613 |
| Random forest | 0.812 | **0.844** | **0.745** | **0.811** | **0.176** | **0.876** |
| Gradient boosting | **0.852** | 0.821 | 0.733 | 0.806 | 0.337 | 0.815 |
| SVM | 0.834 | 0.837 | 0.701 | 0.725 | 0.302 | 0.818 |

In what follows, we aim to compare several link embedding metrics (see Table 2) for the best machine learning model. To evaluate our approach for the first task on Scopus dataset, we can see in Table 4 that suggested by authors new link embedding outperforms existing approaches by all the binary classification quality metrics.

As for the second task, we evaluate LP task over HSE and Scopus datasets in terms of predictive models and link embeddings. In Tables 5 and 6, we could see that the SVM model outperforms the other model on the HSE dataset but Random Forest gets the best AUC for the Scopus dataset, being competitive with the XGBoost and SVM models (SVM performs slightly better). This could happen due to sparse data in Scopus publications after removing 50% link information and lack of ensemble methods for choosing proper negative sampling for such a sparse dataset.

In Tables 7 and 8, we could see that the suggested local proximity operator for link embedding that we call Neighbor Weighted-$L_2$ link embedding outperforms all the other approaches for embedding edges based on node vector representations.

To choose the best predictive model and edge embedding operator, we consider several feature space combination based on List 1. The results of their comparison are shown in Table 9 for the first task and in Table 10 for the second task of LP (the original results appear in *Makarov et al. (2018b)*).

**Table 7 Comparing link embeddings for the link prediction problem on the HSE dataset.**

|  | Precision | Accuracy | F1-score (macro) | F1-score (micro) | Log-loss | ROC–AUC |
|---|---|---|---|---|---|---|
| Average | 0.628 | 0.611 | 0.693 | 0.738 | 1.092 | 0.641 |
| Hadamard | 0.721 | 0.728 | 0.673 | 0.687 | 0.703 | 0.771 |
| Weighted-$L_1$ | 0.776 | 0.765 | 0.727 | 0.759 | 0.376 | 0.817 |
| Weighted-$L_2$ | 0.786 | 0.782 | 0.751 | 0.782 | 0.355 | 0.827 |
| Neighbor Weighted-$L_1$ | 0.816 | 0.834 | 0.762 | 0.801 | 0.214 | 0.839 |
| Neighbor Weighted-$L_2$ | **0.836** | **0.871** | **0.774** | **0.831** | **0.193** | **0.888** |

**Table 8 Comparing link embeddings for the link prediction problem on the Scopus dataset.**

|  | Precision | Accuracy | F1-score (macro) | F1-score (micro) | Log-loss | ROC–AUC |
|---|---|---|---|---|---|---|
| Average | 0.588 | 0.531 | 0.563 | 0.569 | 1.321 | 0.553 |
| Hadamard | 0.672 | 0.637 | 0.611 | 0.632 | 0.784 | 0.626 |
| Weighted-$L_1$ | 0.746 | 0.653 | 0.675 | 0.694 | 0.693 | 0.668 |
| Weighted-$L_2$ | 0.786 | 0.771 | 0.705 | 0.707 | 0.597 | 0.774 |
| Neighbor Weighted-$L_1$ | 0.794 | 0.821 | 0.732 | 0.781 | 0.337 | 0.832 |
| Neighbor Weighted-$L_2$ | **0.812** | **0.844** | **0.745** | **0.811** | **0.176** | **0.876** |

**Table 9 Prediction of publications for 2017 year based on ≤2016 years information.**

|  | Precision | Accuracy | F1-score (macro) | F1-score (micro) | Log-loss | ROC–AUC |
|---|---|---|---|---|---|---|
| (1) | 0.712 | 0.784 | 0.761 | 0.754 | 0.634 | 0.816 |
| (3) | 0.652 | 0.745 | 0.731 | 0.719 | 0.682 | 0.767 |
| (1)+(2) | 0.735 | 0.822 | 0.775 | 0.759 | 0.593 | 0.836 |
| (1)+(3) | 0.728 | 0.796 | 0.781 | **0.789** | 0.618 | 0.827 |
| (1)+(4) | 0.722 | 0.791 | 0.772 | 0.751 | 0.611 | 0.819 |
| (3)+(4) | 0.683 | 0.762 | 0.782 | 0.781 | 0.671 | 0.762 |
| (1)+(2)+(3) | 0.742 | 0.863 | 0.787 | 0.751 | 0.582 | 0.855 |
| (1)+(2)+(4) | 0.738 | 0.852 | 0.791 | 0.731 | 0.604 | 0.842 |
| (1)+(2)+(3)+(4) | **0.798** | **0.866** | **0.793** | 0.786 | **0.573** | **0.878** |

We could see that adding embedding of author research interests as well as the author embedding itself play significant role in improving prediction quality for both tasks. When considering only structural embedding or node similarity features, we obtained worse results in terms of all binary classification quality metrics. In both tasks, the combined approach with direct node similarity scores does not improve the quality of prediction overfitting the model on particular properties thus influencing the predictions for network with missing links. *Makarov et al. (2018a)* evaluate their recommender system on including research interests based only on subject categories from the respective journals index in Scopus. It leads to worse results for LP for the authors with small number of research publication, so they succeeded only predicting so-called strong connections for authors writing at least three to five papers in co-authorship. Our approach

**Table 10** Link prediction for 2017 year on the Scopus dataset.

|  | Precision | Accuracy | F1-score (macro) | F1-score (micro) | Log-loss | ROC–AUC |
|---|---|---|---|---|---|---|
| (1) | 0.767 | 0.831 | 0.703 | 0.783 | 0.236 | 0.859 |
| (3) | 0.731 | 0.822 | 0.698 | 0.778 | 0.244 | 0.813 |
| (1)+(2) | 0.811 | 0.843 | 0.726 | 0.805 | 0.216 | 0.864 |
| (1)+(3) | 0.763 | 0.821 | 0.703 | 0.775 | 0.231 | 0.839 |
| (1)+(4) | 0.772 | 0.833 | 0.715 | 0.794 | 0.223 | 0.861 |
| (3)+(4) | 0.747 | 0.824 | 0.719 | 0.791 | 0.241 | 0.825 |
| (1)+(2)+(3) | 0.808 | 0.835 | 0.724 | 0.807 | 0.193 | 0.866 |
| (1)+(2)+(4) | **0.821** | 0.842 | 0.733 | **0.813** | 0.203 | 0.872 |
| (1)+(2)+(3)+(4) | 0.812 | **0.844** | **0.745** | 0.811 | **0.176** | **0.876** |

allows to work with arbitrary research interest representation thus making it possible to use recommender system for novice researchers with small or zero connections in the network.

## EXPERIMENTS FOR A LARGE NETWORK

To evaluate scalability of our results, we considered the LP task for the large network called AMiner (*Tang et al., 2008*). This network describes collaborations among the authors and contains 1,560,640 nodes and 4,258,615 edges.

For constructing node embeddings we used node2vec with model parameters $p, q = (1,1)$, embedding dimension $d = 128$, length of walk per node equaled $l = 60$, and number of walks per node equaled $n = 3$. We decreased values of $l$ and $n$ in comparison with default values, because our computer terminated process due to memory issues.

We studied the impact of train/test split on different edge embeddings operators while fixing Logistic Regression model for LP. We considered train set, consisting of 20%, 40%, 60%, 80% of the graph edges while averaging binary classification quality metrics over five negative sampling providing negative examples for non-existent edges. We compared the Log-loss (Figs. 2A and 2B) and Accuracy (Figs. 2C and 2D) metrics computed for train and test sets using different edge embeddings. As a result, we obtained that Hadamard and Neighbor Weighted-$L_2$ edge embedding operators represent highly accurate results while trained on sparse data from the original graph. Moreover, increasing the size of train data (which is the case for our temporal co-authorship network) The Hadamard product becomes inferior to our Neighbor Weighted-$L_2$ operator. In addition, the Neighbor Weighted-$L_1$ operator also showed greater performance in contrast to the HSE dataset.

It was interesting to find out that overall performance of node2vec node embedding model produces very precise results for LP task, which may vary depending on graph as was shown in *Grover & Leskovec (2016)*.

## DISCUSSION

We have obtained that combined approach of embedding co-authorship and keywords co-occurrence networks, while preserving several author attributes leads to significant improvement in the classification quality metrics for predicting future links and LP tasks. However, we will continue to experiment with node2vec model. The (1) feature space

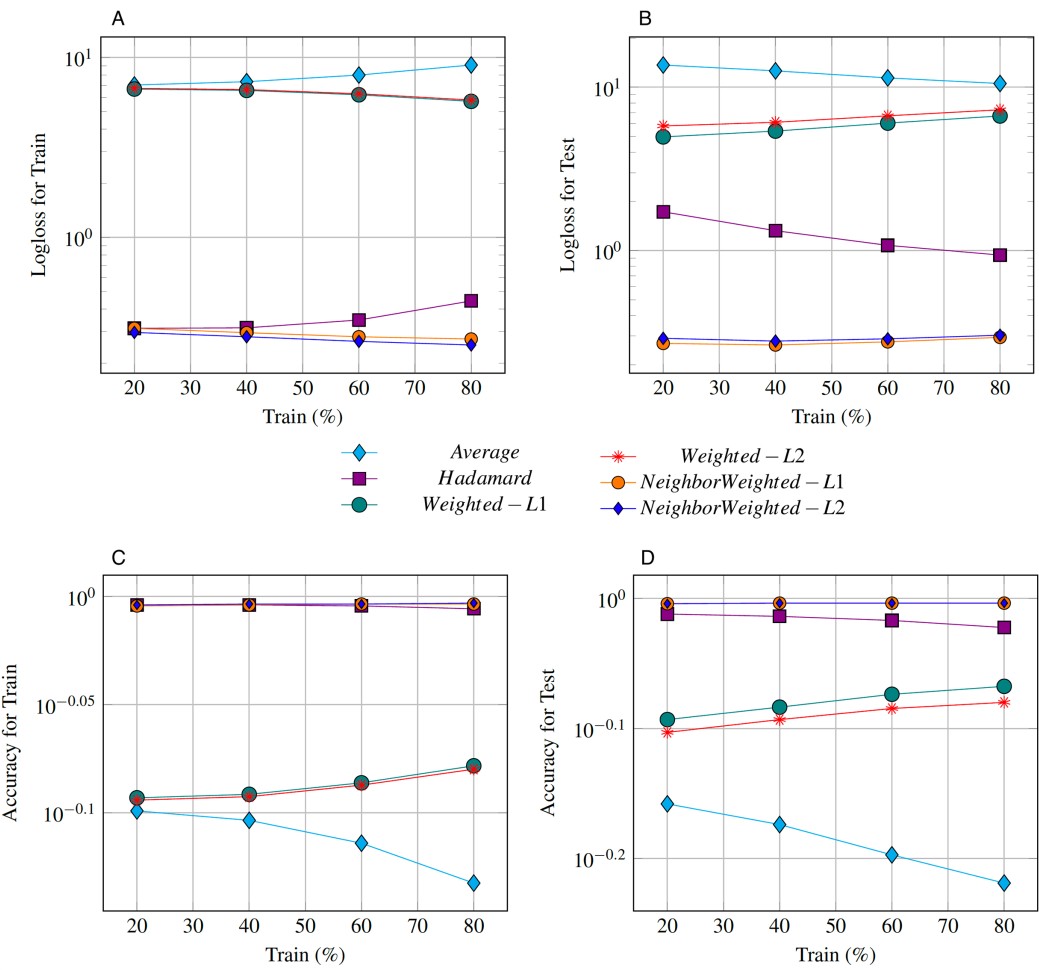

**Figure 2 Low log-loss and high accuracy (plotted log-scale Y-axis) represent the best edge embedding.** We show that Neighbor Weighted-$L_1$ and Neighbor Weighted-$L_2$ operators are on par with the state-of-art Hadamard product presented in *Grover & Leskovec (2016)* and outperform it when train data size increases based on Log-loss (A and B) and Accuracy (C and D) metrics computed for train and test sets.  

from the List 1 is node2vec model with $d = 128$ (compared also on $d = 128$) and parameters $p, q = (1, 0.25)$ obtained by us via model fitting on a given $d$ and logistic regression baseline model. The small $q$ value shows that considering local proximity was more important than proximities of highest orders.

However, we aim to further study this question taking into account modern methods of graph auto-encoders (*Kipf & Welling, 2016*). We are further working on a new embedding model called JONNE learning high quality node representations in tandem with edge representations. We proved the embeddings learned by JONNEE is almost always superior then of those learned by state-of-the-art models of all different types: matrix factorization based, sequence-based and deep learning based but the model has a certain drawback of longer training similar to matrix factorizations but less parallelizable, thus giving us the dilemma to choose between quality and processing speed of suggested solution for edge embedding construction.

While the LP task still remains hard problem for network analysis, its application to matching collaborators based on structural, attribute and content information shows promising results on applicability of graph-based recommender system predicting links in the co-authorship network and incorporating author research interests in collaborative patterns. We aim to generalize the model based on full-text extraction of research interests from collections of source documents and studying deep learning solutions for representing combined embedding of structural and content information for co-authorship networks.

The code for computing all the models with respect to classification evaluation, choosing proper edge embedding operator and tuning hyper-parameters of node embeddings will be uploaded on GitHub (http://github.com/makarovia/jcdl2018/) including the HSE and Scopus datasets.

## CONCLUSION

We have improved recommender systems (*Makarov, Bulanov & Zhukov, 2017*; *Makarov et al., 2018a*) based on choosing proper link embedding operator (*Makarov et al., 2018b*) and including research interest information presented as embedding of nodes in keywords co-occurrence network connecting keywords relating to a given research article. We have compared several machine learning models for future and missing LP problems interpreted as a binary classification problem. The edge embedding operator suggested by *Makarov et al. (2018b)* edge embedding operator called Neighbor Weighted-$L_2$ (see Table 2) outperforms all the other edge embedding functions due to involving the neighborhood of the edge in the graph and was properly evaluated in this paper for both tasks. Among the machine learning models, SVM outperforms all the others except the LP problem on the sparse Scopus dataset while XGBoost was significantly overfitted; however, training SVM for large graphs is computationally hard.

Such a constructed model may be considered as a recommender system for searching collaborators based on mutual research interests and publishing patterns.
The recommender system demonstrates good results on predicting new collaborations between existing authors even if they have small number of data in co-authorship network due to availability of their research interests.

We are looking forward to the evaluation of our system for universities, who have to deal with the problems of finding an expert based on text for evaluation, matchmaking for co-authored research papers with novice researchers, searching for collaborators on specific grant proposal or proper scientific advisers.

Focusing only on machine learning task is not suitable for real-world application involving social interactions, so we aim to implement framework with the possibility to manually add positive and negative preferences for collaboration recommendations, thus providing a useful service which could be integrated in university business process of managing researchers' publication activity.

Our system may also be used for predicting the number of publications corresponding to a given administrative staff unit using network collaborative patterns and thus able to evaluate the efficiency of the authors or the whole staff. It also may be used for

suggesting collaborations between separate staff units by considering combined network with staff units as vertices and weighted by the number of publications mutual connections between them. Evaluating such a network for HSE University give us the picture that most popular faculties of Economics and Management have a lot of mutual connections due to many researchers working at these faculties, but limiting these connection only on Scopus publication leads to the influence of Computer Science and Engineering faculties which showed trend of computer science research in applied sciences. We leave for the future work consideration of the applicability of our system which suggests a new university publication strategy based on collaboration patterns and invites researchers to compare the existing solutions on the HSE researchers' dataset.

## ACKNOWLEDGEMENTS

A part of the article is extended and revised version of presented oral talk at international conference AIST'18 and posters at ACM/IEEE JCDL'18, WebSci'18 and SunBelt'18. We thank all the colleagues from NRU HSE participated in discussion of this research.

### Funding

The work was supported by the Russian Science Foundation under grant 17-11-01294 and performed at the National Research University Higher School of Economics, Russia. The funders had no role in study design, data collection and analysis, decision to publish, or preparation of the manuscript.

### Grant Disclosures

The following grant information was disclosed by the authors:
Russian Science Foundation: 17-11-01294.
National Research University Higher School of Economics, Russia.

### Competing Interests

The authors declare that they have no competing interests.

### Author Contributions

- Ilya Makarov conceived and designed the experiments, analyzed the data, prepared figures and/or tables, performed the computation work, authored or reviewed drafts of the paper, approved the final draft.
- Olga Gerasimova performed the experiments, analyzed the data, prepared figures and/or tables, performed the computation work, approved the final draft.
- Pavel Sulimov performed the experiments, analyzed the data, prepared figures and/or tables, performed the computation work, approved the final draft.
- Leonid E. Zhukov conceived and designed the experiments, analyzed the data, performed the computation work, approved the final draft.

## Data Availability

GitHub: https://github.com/MakarovIA/jcdl2018.

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
