# Peer review of "Dual network embedding for representing research interests in the link prediction problem on co-authorship networks"

_PeerJ Computer Science, doi:10.7717/peerj-cs.172_

## Round 0.1 · original submission · Major Revisions

Thank you for submitting your manuscript to PeerJ. After careful consideration, we feel that it has merit but does not fully meet PeerJ’s publication criteria as it currently stands. Therefore, we invite you to submit a revised version of the manuscript that addresses the points raised during the review process.

Reviewer 1 ·

Basic reporting

The structure of the paper is correct. However, the text presents many grammatical and structural errors that make it hard to follow, the English is far from perfection.
Specific comments:
1. It is noted that your manuscript needs careful editing paying particular attention to English grammar, spelling, and sentence structure so that the goals and results of study are clear to the reader.
 Page 1 line 12: the sentence “We compare existing graph features engineering…”should be improved to better express its meaning.
 Page 1 line 25: the sentence “the authors measuring their influence on…”should be improved to better express its meaning.
 There are misusing between singular and plural. Like Page 1 line 26: “such methods does not” supposed to be “such method does not”. There are many cases like this throughout the text.
 Page 1 line 28: “for e.g.,” is not correct. You can say “for example,” or “e.g.,”.
 Page 2 line 88: “the theory of hidden representations has impacted on…” reads better. The same as page 3 line 106 “Several works cover the node attributes…”, page 3 line 114 “…space of author’s research interests…”, and page 3 line 115 “Based on this information…”.
 Page 3 line 128: The term “author homonimy” is unusual in the literature. The same page line 132 “common neighbors” is a metric not measure.
 Page 4 line 160: What do you mean by the expression "adjust the feature vector based on …"?
 Page 4 line 176: “egde” should be “edge”.
 Page 9 line 289: The incomplete sentence “The applicability of our system for …”.
2. There are some references suggestions follows:
 Some important recent references are missing, e.g. VOPRec: Vector Representation Learning of Papers with Text Information and Structural Identity for Recommendation, IEEE Transactions on Emerging Topics in Computing, 2018. DOI: 10.1109/TETC.2018.2830698
 Page 1 line 28: The review [48] is a little out-of-date.
 Page 1 line 43: Please check the reference [10].
 Avoid using a reference as part of the sentence by replacing it by the authors' name. For example, page 2 line 70: “In [13], the authors presented…” the sentence should be structured as “Gao et al. [13] presented…”. here are many cases like this throughout the text.
 Page 3 line 102: The wrong format of reference “[Carstens et al.]”.
 What is the different between the reference [23] and [24]?

Experimental design

The combination of these features shows the improvement of recommendation.

Validity of the findings

The authors studied the link prediction problem considering network structure and research interests based on network embedding and manual features in the co-authorship network.

Additional comments

3. The keyword “Co-occurrence network” is suggested to appear in the abstract part for readers to understand the key-word co-occurrence network in the following text.
4. Page 3 line 138: What’s the relation between the description of “the influential persons” and the datasets?
5. According to the page 3 line 132 the author disambiguation is based on three rules, however the author disambiguation in line 146 is solved by common neighbors. The rules of author disambiguation need to be clearer. How to perform the author disambiguation on these datasets?
6. Page 5 line 175: What are “specific component-wise functions”? It supposed to explain in the text clearly.
7. Page 6 line 210: What are “machine learning models”? Are they Logistic Regression with Lasso regularization, Random Forest, Gradient Boosting, and SVM in the following text? The methods description need to be earlier in line 210.
8. Page 6 line 212: What do you mean by “based on List 5”?
9. Page 7 line 255: What is the “2.6” in sentence “The suggested by authors in [29] edge embedding from 2.6…”? More explanation need to be given.

Reviewer 2 ·

Basic reporting

The authors propose a method for link prediction in co-authorship based on modern machine learning techniques applied to the network itself combined with information from features of keywords derived from publications.

The text is clear but needs some improvements, I recommend a carefully revised to achieve the quality of the journal. For instance, line 151 and the repetitions of construction We ... (see section 4).

The references cover the most important and related works. However, because peerj is a multidisciplinary journal and this article may be of interest to scientists without a strong background on machine learning, some abbreviations and terms should be explained or at least referenced. For instance: AUC, F1-SCORE (MACRO/MICRO), LOGLOSS, ROC-AUC, XGBoost, SVM.

The proposed problem is clear and very relevant to the area.

Experimental design

The analysis is solid and the results confirms that there is a considerable improvement on the prediction quality by combining topological information of the co-authorship networks with contextual information from keywords.

The authors provide a set of codes to replicate the findings.

Validity of the findings

Since the employed datasets cover only one institution, the model may be finding collaborations related to a few grants projects or among researchers in the same group. I think a discussion about this specific characteristic should be added to the paper.

As a continuation of the previous point, the method was applied to a very small dataset of authors. However, much larger and partially disambiguated datasets are now available, such as the Microsoft Academic Graph. I'm very curious to see if the proposed technique scales well for a larger and broader dataset. A simple analysis that could be performed would be sampling a fraction of the current datasets (e.g. 20%, 40%, 60%, 80%) and check the classification performance (just one metric is enough to illustrate the effects of the data size).

The method used for evaluating the classifiers needs to be described or at least referenced. Also, if you are using a n-fold cross validation, why there is no deviations for the performance metrics?

There is no information of the number of authors and publications for the considered Scopus subset.

---

## Round 0.2 · accepted · Accept

Your article has been re-reviewed and is accepted.

Reviewer 1 ·

Basic reporting

no comment

Experimental design

no comment

Validity of the findings

no comment

Additional comments

All my concerns have been solved.

Reviewer 2 ·

Basic reporting

no comment

Experimental design

no comment

Validity of the findings

no comment

Additional comments

The authors have revised the manuscript according to the reviewer comments. The paper can be accepted now.